

# Thermal Regime of High Arctic Tundra Ponds, Nanuit Itillinga (Polar Bear Pass), Nunavut, Canada

Kathy L. Young[1], Laura C. Brown[2]

[1]Faculty of Environmental & Urban Change, York University, Toronto, Ontario, M9C 4A4, CANADA

[2]Department of Geography, Geomatics and Environment, University of Toronto, Mississauga, L5L 1C6, CANADA

*Correspondence to: Laura C. Brown (lc.brown@utoronto.ca)*

**Abstract.** This study evaluates the seasonal and inter-seasonal temperature regime of small tundra ponds ubiquitous to an extensive low-gradient wetland in the Canadian High Arctic. Pond temperatures can modify evaporation and ground thaw rates, losses of greenhouse gases and control the timing and emergence of insects and larvae critical for migratory bird feeding habits. We focus our study on thaw ponds with a range of hydrologic linkages and sizes across Nanuit Itillinga, formerly known as Polar Bear Pass (PBP), Bathurst Island, and whenever possible, compare their thermal signals to other Arctic ponds. Pond temperatures and water levels were evaluated using temperature water level loggers and verified by regular manual measurements. Other environmental data collected included microclimate, frost table depths and water conductivity. Our results show that there is much variability in pond thermal regimes over seasons, years, and space. Cumulative relative pond temperatures were similar across years, with ponds normally reaching 10-15° C for short to longer periods except in 2013, a cold summer season when pond temperatures never exceeded 5º C. Pond frost tables and water conductivities respond to variable substrate conditions and pond thermal patterns. This study contributes to the ongoing discussion of climate warming and its impact on Arctic landscapes.

## 1 Introduction

Arctic landscapes are warming faster than temperate locations (e.g., Linderholm et al., 2018; Sim et al., 2019; Kreplin et al., 2021; McCrystall et al., 2021; Webb et al., 2022), specifically, up to four times faster than the globe since 1979 (Rantanen et al., 2022). This rapid warming has implications for thawing of permafrost, the alteration of hydrologic regimes (snow on/off, ice-free duration, runoff), changes in hydrologic pathways, and a suite of other environmental and ecosystem impacts (e.g., Webb et al., 2022). There is an expectation that the Arctic will transition from a snow-dominated to a rainfall-dominated regime, especially with an occurrence of higher rainfall in the Fall (McCrystall et al., 2021), and some suggest that it will occur earlier than initially modelled (McCrystall et al., 2021).

Recently, there has also been a suite of studies and literature reviews evaluating the warming of water bodies (ponds, lakes, streams) across arctic landscapes (e.g., Dranga et al., 2017; Lehnherr et al., 2018; McEwen and Butler, 2018; Saros et al., 2023) and elsewhere (e.g., small ponds in Alaska — Andresen and Lougheed, 2015; wetlands in the high Andes — Dangles et al., 2017; ponds/lakes in W. Greenland — Higgens et al., 2019). Saros et al. (2023) remark that "knowing temperatures and





thermal structure within lakes and flowing waters at present and predicting their changes in the future is critical for
understanding how aquatic ecosystems will undergo future changes". Others, particularly in the subarctic, are concerned with
deepening talik development (unfrozen ground) and lake expansion due to the interactions of shoreline erosion, rising lake
bottom temperatures and the occurrence of deep snow near lake/pond shorelines (Roy-Leveillee and Burn, 2017).
In the Eastern Canadian Arctic, warming has been especially prominent since 2000, with growing loss of glacier ice,
permafrost thaw, disappearance of late-lying snowbeds and the drying of small, patchy wetlands (Woo and Young, 2014).
There has been a growing interest in how climate warming will affect northern wetlands here, as temperature increases can
enhance evaporation rates, thaw permafrost, drain ponds, or initiate the development of new ponds. Warmer substrates can
increase the thaw depth of water bodies, and temperature has an impact on vegetation growth, greenhouse gases, including
water vapour, methane and carbon dioxide (Negandhi et al., 2013; Andresen and Lougheed, 2015; Wrona et al., 2016; Zandt
et al., 2020; Kreplin et al., 2021; Dyke and Sladen, 2022, Miner et al., 2022; Rheder et al., 2023).
This study of pond thermal regimes at Nanuit Itillinga (PBP) adds to this body of literature by evaluating the seasonal and
inter-seasonal temperature regime of small tundra ponds ubiquitous to an extensive low-gradient wetland in the Canadian High
Arctic spanning warm and cool spring/summers.
First, we examine the spatial and temporal variability of pond temperatures across PBP, highlighting variations in
pond location, hydrological linkages, and response to climate variability on a seasonal and annual basis. We then evaluate
whether July average pond temperatures vary across ponds at PBP and in relation to other Arctic sites. This study also examines
the air-pond temperature relationship, and we explore the impact of pond temperatures on the local environment in terms of
pond ground thaw and water chemistry (i.e., specific conductivity), as others have reported deeper thaw in warmer water bodies
and higher nutrient loads (e.g., Saros et al., 2023). Finally, we place our results in context of other studies and discuss how
these ponds are being affected by both cool and warm seasons, and what can be expected in the future for these ponds and the
ecosystems that depend on them.

## 2 Study Area

The study took place at the extensive low-lying wetland, Nanuit Itillinga (Polar Bear Pass-PBP) Bathurst Island, Nunavut (75.
72º N 98.67º W) from 2007 to 2015, with focused pond studies in 2008 and 2009. PBP is a National Wildlife Area and a
Ramsar wetland site of international importance (Baker et al., 2021). The wetland itself is about 20 km long and 5 km wide,
and is boarded by low-lying hills ranging upwards of 160 m to the north and 170 m to the south (Caledonian River, District of
Franklin, NWT, 1985 topographic map (1:50,000), 68H/11, edition 1)). The low-lying wetland encompasses two small lakes
and a myriad of ponds (small to large) exhibiting uniform or irregular shapes (Fig. 1). Detailed characteristics of the PBP
watershed and study ponds can be found elsewhere (e.g., Abnizova, 2013; Abnizova et al., 2014; Young et al., 2017), however
we do provide a summary of the study ponds including location, pond area, water and frost table depths and details on pond
substrate where available (Table 1, Supplementary Figure 1).



**Figure 1: Location of the PBP catchment on Bathurst Island, Nunavut (a,b) and satellite imagery of the eastern and central lowland area with an air photo inset showing the wetland with numerous ponds and upland areas (b,c). Photo image was taken on July 10, 2009. Satellite imagery obtained through EOS Landviewer (eos.com): Sentinel-2 Level2A, Bands 11, 8A, 02, September 6, 2022.**



**Table 1: Summary of pond site location, pond surface area, and water table (WT) (2007-2010). Frost table (FT) refers to the maximum frost table for the season or period specified. More details can be found in Croft (2011) and Abnizova (2013)**

|  | Pond1 | Pond2 | Pond3 | Pond4 | Pond5 | Pond6 | Pond7 | Pond8 | Pond9 | Pond10 |
|---|---|---|---|---|---|---|---|---|---|---|
| Coordinates | 75° 43' 30.50" 98° 25' 53.83" | 75° 43' 30.82" 98° 25' 54.60" | 75° 43' 26.67" 98° 25' 59.99" | 75° 43' 28.71" 98° 25' 33.49" | 75° 43' 27.24" 98° 25' 28.71" | 75° 43' 27.24" 98° 25' 28.71" | 75° 43' 23.72" 98° 25' 26.76" | 75° 43' 27.24" 98° 25' 14.45" | 75° 42' 42.74" 98° 26' 32.93" | 75° 42' 40.50" 98° 26' 27.14" |
| Surface area (m$^2$) | 6375 | 275 | 7975 | 750 | 1850 | 1850 | 10950 | 1000 | 5200 | 6475 |
| **2007** |  |  |  |  |  |  |  |  |  |  |
| WT (mm) | 344 | 415 | 359 | 265 | 243 | 176 | 177 | 68 | 99 | 123 |
| FT (mm) | -635 | -410 | -645 | -475 | -310 | -710 | -650 | -595 | -851 | -432 |
| β* | 0.10 | 0.08 | 0.11 | 0.08 | 0.05 | 0.11 | 0.10 | 0.10 | - | - |
| Soil Colour | – | – | – | – | Gray | Gray | Gray | Gray | Very Dark Grayish Brown | Grayish Brown |
| %Organics | 1.65 | – | – | – | 2.37 | – | 1.49 | 1.9 | 10.9 | 3.42 |
| **2008** |  |  |  |  |  |  |  |  |  |  |
| WT (mm) | 302 | 382 | 347 | 215 | 247 | 107 | 162 | 80 | – | 58 |
| FT (mm) | -731 | -569 | -738 | -544 | -580 | -814 | -745 | -675 | – | -529 |
| β* | 0.11 | 0.07 | 0.10 | 0.07 | 0.07 | 0.10 | 0.09 | 0.08 | – | 0.07 |
| **2009** |  |  |  |  |  |  |  |  |  |  |
| WT (mm) | 348 | 433 | 348 | 236 | 200 | 195 | 171 | 158 | 286 | 103 |
| FT (mm) | -689 | -489 | -644 | -534 | -524 | -707 | -680 | -631 | -408 | -659 |
| β* | 0.08 | 0.06 | 0.08 | 0.06 | 0.06 | 0.09 | 0.08 | 0.08 | – | 0.08 |
| Bulk Density (g/cm$^3$) | 1.87 | 2.10 | 2.27 | 1.83 | – | 1.82 | – | 1.81 | – | 2.28 |
| Porosity (%) | 22 | 25 | 17 | 25 | – | 14 | – | 13 | – | 11 |
| **2010** |  |  |  |  |  |  |  |  |  |  |
| WT (mm) | 334 | 456 | 373 | 270 | 266 | 152 | 195 | 178 | 275 | 114 |
| FT (mm) | -649 | -502 | -632 | -492 | -512 | -702 | -678 | -529 | -334 | -428 |
| β* | 0.10 | 0.08 | 0.09 | 0.07 | 0.08 | 0.11 | 0.10 | 0.10 | – | 0.07 |





Table 1-continued

|  | Pond11 | Pond12 | Pond13 | Creek Pond1 | Creek Pond2 | Meadow Pond | Pingo Pond |
|---|---|---|---|---|---|---|---|
| Coordinates | 75° 43' 33.86" 98° 24' 8.66" | 75°43' 26.91" 98°22' 59.49" | 75°43' 26.91" 98°22' 59.49" | 75°43' 23.08" 98°26' 33.74" | 75° 43' 22.08" 98° 22' 50.65" | 75°43' 34.92" 98°31' 2.89" | 75° 43' 36.14" 98° 31' 29.93" |
| Surface area (m²) | 550 | 10475 | 1725 | 650 | 1250 | 10666 | 726 |
| **2007** | | | | | | | |
| WT (mm) | 132 | – | – | – | – | – | – |
| FT (mm) | -555 | – | – | – | – | – | – |
| Pond Soil Colour | Light Brownish Gray | Olive Gray | Dark Gray | – | – | – | – |
| %Organics | 5.54 | 3.03 | 2.92 | – | – | – | – |
| **2008** | | | | | | | |
| WT (mm) | 141 | 168 | 236 | 202 | 196 | 273 | 149 |
| FT (mm) | -863 | -951 | -819 | -550 | -698 | -388 | -582 |
| $\beta$* | 0.09 | 0.12 | 0.11 | 0.08 | 0.09 | 0.06 | 0.06 |
| **2009** | | | | | | | |
| WT (mm) | 243 | 256 | 303 | 261 | 215 | 525 | 380 |
| FT (mm) | -825 | -903 | -882 | -544 | -612 | -307 | -252 |
| $\beta$* | 0.10 | 0.11 | 0.10 | 0.07 | 0.08 | 0.04 | 0.03 |
| Bulk Density (g/cm³ | 1.01 | 1.53 | 1.83 | 1.87 | 2.31 | – | – |
| Porosity (%) | 15 | 26 | 13 | 45 | 16 | – | – |
| **2010** | | | | | | | |
| WT (mm) | 247 | 193 | 254 | 328 | 172 | 525 | 435 |
| FT (mm) | -810 | -661 | -534 | -502 | -609 | -294 | -211 |
| $\beta$* | 0.13 | | | 0.07 | 0.09 | 0.05 | 0.03 |










Table 1-Continued

|  | East Small Pond | East Med. Pond | East Large Pond | South Small Pond | South Med. Pond | South Large Pond | West Small Pond | West Med. Pond | West Large Pond |
|---|---|---|---|---|---|---|---|---|---|
| Coordinates | 75° 44' 6.01" 98° 5' 42.73" | 75° 44' 4.80" 98° 5' 28.15" | 75° 44' 10.01" 98° 5' 31.62" | 75° 41' 50.93" 98° 20' 49.33" | 75° 41' 51.36" 98° 20' 40.83" | 75° 41' 47.19" 98° 20' 28.74" | 75° 42' 20.53" 98° 45' 18.70" | 75° 42' 25.24" 98° 45' 17.19" | 75° 42' 21.47" 98° 45' 29.00" |
| Surface area ($m^2$) | 100 | 1500 | 24500 | 350 | 2075 | 3575 | 2550 | 7050 | 31400 |
| **2007** (July 26-Aug. 2) |  |  |  |  |  |  |  |  |  |
| WT (mm) | 70 | 110 | 125 | – | – | – | 60 | 100 | 130 |
| FT (mm) | -460 | -740 | -760 | – | – | – | -885 | -1088 | -1020 |
| Pond Soil Colour | Gray | Gray | Light Brownish Gray | – | – | – | – | Light Olive Gray | – |
| % Organics | 3.26 | 5.1 | 1.06 | – | – | – | – | 1.42 | – |
| **2008** (Aug. 26-Sept 1-fall) |  |  |  |  |  |  |  |  |  |
| WT (mm) | 135 | 222 | 275 | 180 | 400 | 328 | 261 | 150 | 262 |
| FT (mm) | -556 | -648 | -588 | -675 | -653 | -443 | -527 | -852 | -893 |
| **2009** (Aug. 3-5) Summer |  |  |  |  |  |  |  |  |  |
| WT (mm) | 270 | 290 | 245 | 410 | 330 | 381 | 340 | 268 | 350 |
| FT (mm) | -921 | -603 | -625 | -386 | -323 | -492 | -677 | -713 | -876 |
| **2009** (Aug. 26-30)-Fall |  |  |  |  |  |  |  |  |  |
| WT (mm) | 260 | 294 | 223 | – | – | – | 406 | 222 | 446 |
| FT (mm) | -968 | -948 | -1020 | – | – | – | -618 | -959 | -798 |
| **2010** (June 19-24)-Spring |  |  |  |  |  |  |  |  |  |
| WT (mm) | 198 | 165 | 104 | 425 | 456 | 284 | 228 | 188 | 195 |
| FT (mm) | -15 | -23 | -20 | 0 | -6 | -30 | -75 | -68 | -30 |






*$\beta$, a coefficient is determined after Woo (1983), where $Z_f = \beta(t^{0.5})$. Here $Z_f$ is the depth to the frost table (taken here as the
Max FT), $t$ is time in days after the initial thaw period. Others (e.g., Qingbai et al., 2015) have referred to $\beta$ as the *Edaphic*
*Factor* or *scaling parameter*, which considers soil characteristics such as thermal conductivity, bulk density, soil water
content, and latent heat of fusion. Qingbai et al. (2015) also used cumulative degree-days for $t$ instead of days since ground
thaw.

**3 Methods**
In this paper, we focus on describing the thermal regime of selected ponds (small to large) across PBP; ones centrally located,
and others situated at the eastern, western, and southern edges of the wetland. Some ponds are isolated (no apparent water
sources feeding into them), while other ponds are connected or temporarily connected to small creeks, other ponds, and/or
meltwater from nearby late-lying snowbeds (Abnizova et al., 2014).  Ponds at PBP are defined as having a maximum water
depth of < 2.0 m (National Wetland Working Group, 1997; Woo, 2012). Substrate type varied across the ponds: some were
firm, light-coloured, while others had rocky pond bottoms or dark, soft organic beds (see Table 1, Supplementary Figure 1).
Water levels in the ponds were continuously monitored with Ecotone water level recorders (Remote Data Systems Inc., ± 2.54
mm) or with HOBO water level sensors (± 3 mm), which also measured temperature (± 0.2º C) on an hourly basis. These
sensors were typically placed in the centre of the pond in perforated, screened water wells (5.1 to 7.6 cm dia.), open to the
atmosphere and secured into frozen ground. In ponds which were difficult to access due to a soft substrate, water wells and
HOBO sensors were placed near the shoreline, and occasionally these sensors were placed on the pond bed, especially in
remote ponds. HOBO water level sensors were also placed in dry wells dug into wet meadow areas to monitor the atmospheric
pressure. The difference in the pressure determined by HOBO water level sensors in the ponds and the atmosphere allowed
pond water levels (mm) to be derived (see Rosenberry and Hayashi, 2013). Like other shallow arctic ponds, the study ponds
were generally well-mixed negating any concern about significant differences in temperatures of the bed and the water column
(Dyke and Sladen, 2022).

111        Manual measurements of water levels at centrally located ponds were made on a regular basis, usually every one to

two days, with a measuring tape (± 5 mm) to verify continuous measurements. Less frequent manual measurements while
manual estimates were made at distant ponds. Small HOBO temperature sensors (± 0.1º C) were also deployed in ponds to
track hourly temperatures.  Manual estimates of temperature and conductivity were made with a YSI conductivity meter (±
0.2º C, ±1 µS/cm), while a Hanna pH meter provided an additional check on temperature (± 0.2º C) along with water pH (±
0.2). In 2008, hourly estimates over several days were made of water chemistry in the selected ponds, including water level
(m), temperature (± 0.15º C), conductivity (± 1.0 µS/cm), dissolved oxygen (%), and pH (± 0.2), using a YSI 600
Multiparameter Sonde (Abnizova et al., 2014).  In 2009, the YSI sonde was also used to obtain manual estimates of water
chemistry in several centrally located ponds, and in 2012 pH and conductivity were monitored in Pond 1 on an hourly interval
over several days.


121       To examine generalized patterns of spatial and temporal trends between air and pond temperatures, Pearson

correlations (r) of pond to air temperatures (2008, 2009) were determined for daily mean temperatures $\geq 0°$ C, avoiding the
flattening of data at low and high temperatures (Johnson et al., 2014).  To investigate inter-site correlations, water temperatures
at Pond 1 were evaluated in relation to other ponds. Prior to correlation analysis, air and pond temperature data were checked
for normalcy using the Shapiro Wilk test, $\alpha = 0.05$. Like others (e.g., Johnson et al., 2014), the Durbin-Watson test was used
as a diagnostic of autocorrelation in regression model residuals. Given that autocorrelation did exist amongst the data (k-1)
and is commonly found when comparing air to water temperatures (Johnson et al., 2014), no further work was carried out to
develop a predictive model between air temperature and pond water.  A Student-t test ($\alpha = 0.05$) was used to compare means
of pond water temperature (Tw) when appropriate in the study (Bluman, 2006).
Changes in ground thaw can alter drainage patterns and water storage in ponds (Young and Woo, 2003; Rehder et al., 2023).
Thaw depth in ponds was measured by probing the ground with a metal rod (± 10 mm) twice a week near water wells early in
the season and then weekly once ground thaw slowed, providing a means of assessing active layer development and re-freeze
(i.e., 2008, 2009) (Abnizova et al., 2014). Climate data (e.g., air temperature) were obtained from the main automatic weather
station located near the PBP cabin situated on the plateau above the wetland (see Young and Labine, 2010; Young et al., 2013;
Miller and Young, 2016 for additional details on instrumentation, sensor siting, and frequency of monitoring). These data
allowed us to examine the air temperature-pond response, and to place our results in context of the variable spring and summer
conditions over several years at PBP, and in relation to the nearest government weather station at Resolute Bay (Qausuittuq),
Cornwallis Island, Nunavut (74.72° N 94.97° W) about 146 km to the southwest.

## 139  4 Results

### 140  4.1 Seasonal thermal regime

The seasonal regime of ponds at PBP over a number of years was explored using the detailed data from Pond 1 (Fig. 2: 2007-
2015). It is a medium-sized pond downslope of a wet meadow and lingering deep snowbed located in the lee of a hillslope
(Young et al., 2017). The inset diagram illustrates the range of pond temperatures during two extreme years (2012-warm vs.
2013-cool).  In warm years (e.g., 2010, 2012), pond water temperatures warm rapidly by mid-June with peak pond temperatures
reaching about 15° C for extended periods. There is considerable variability from year-to-year, but the general trend is warming
in June, elevated temperatures in July and then falling temperatures in August. In context of Resolute air temperatures from
1948 to 2015 (Environment Canada (Resolute): weather.gc.ca), the 2012 JJA air temperature (4.6° C) was the 2nd highest on
record, while the 2013 JJA average (0.9° C) was one of the coolest (ranked 11th coolest, along with three other years) over a
period of 68 years.  The anomalously warm summer 2012 and cool summer 2013 were a result of opposing NAO phases. The
summer of 2012 had a very negative NAO (resulting in warm temperatures and substantial snow and sea ice melt)  (Overland
et al., 2013), while the summer of 2013 had a positive NAO with anomalously low spring and summer temperatures through
the Canadian North, dipping 1-3°C cooler in the High Arctic relative to 2007 – 2012 (Overland et al., 2014).





Figure 3 shows a similar response between Pond 1 and the centrally located ponds. Pond temperatures (Tw) are
generally higher than the air temperature during the thaw season when temperatures are > 0º C, with most also trending the air
temperature signal. This pattern suggests well mixed conditions as noted for shallow ponds elsewhere in the Arctic (Dyke and
Sladen, 2022). In 2009, Pond 1 Tw correlated well with all pond temperatures (r > 0.8 to > 0.9) including ones at the periphery
of the Pass (east, west, and south). In 2008, a similar pattern between Pond 1 and the other ponds across the wetland emerged
(r > 0.8 to > 0.9), except for the Meadow Pond (r (66) = 0.34, p = 0.005). The Meadow Pond is located in a small, elevated
valley, and unlike most ponds, has deep residual snowbeds lying on the adjacent slopes. The nearby Pingo Pond is not fed by
the snowbeds. Meltwater from lingering snowbeds feeding the Meadow Pond ensures that pond temperatures are dampened
in comparison to more exposed and isolated wetland ponds. Others have observed this pattern on northeast Ellesmere Island
(e.g., Smol and Douglas, 2007).



**Figure 2: Seasonal regime of water temperature (Tw) at Pond 1, PBP (2007-2015). Manual measurements for 2007 and daily averages**
**for 2009-2015 are plotted. The inset diagram highlights the variability in Tw for two different seasons: 2012-warm and 2013-cool.**





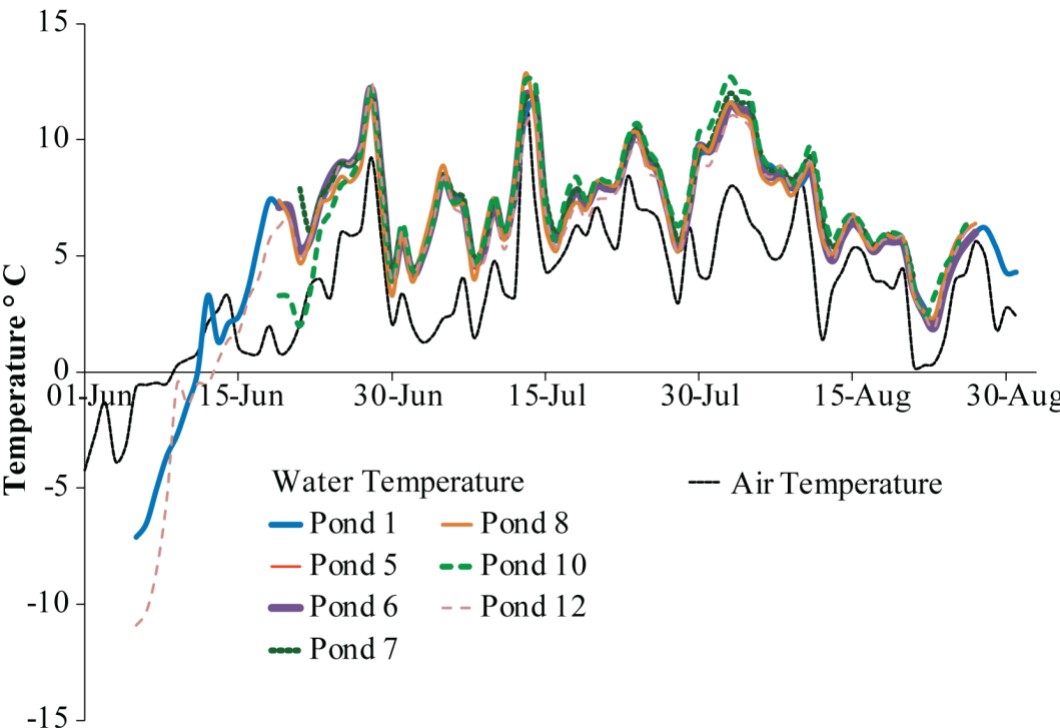

**Figure 3: Daily average water temperature of Pond 1 versus other centrally located ponds at PBP, 2009. The daily average air temperature is also plotted.**

Figure 4 illustrates the seasonal rhythm (2007, 2008, and 2009) of medium-sized ponds across PBP. Here we select Pond 6 in the central cluster of ponds as representative of a medium-sized pond. In 2007, by July 20, Pond 6 was slightly warmer than other two ponds, though the rhythms – peaks and troughs of pond Tw remained similar. Warming is comparable to Fig. 2, where there is some variation in Tw from year-to-year, though these ponds do not exceed 15º C for prolonged periods. Young and Labine (2010) found that environmental conditions were slightly cooler in the eastern part of PBP owing to nearness to the Arctic Ocean, but the largest microclimatic discrepancies across PBP were related to differences in net radiation (surface albedo) and ground heat flux (substrate, vegetation, etc.) (Young et al., 2010). A Student t-test (2-tailed, $\alpha = 0.05$) reveals that in 2008, Pond 6's mean Tw (7.3º C ± 2.3, n = 69) was not significantly different than the East Medium Pond (7.6º C ± 3.5, n=68) but it was for the West Medium Pond (6.6º C ± 4.5, n = 77). In 2009, both the East (5.7º C ± 3.8, n=95) and West Medium (4.3º C ± 5.5, n = 94) ponds had significantly different mean water temperatures than Pond 6 (6.6º C ± 4.5, n=80).




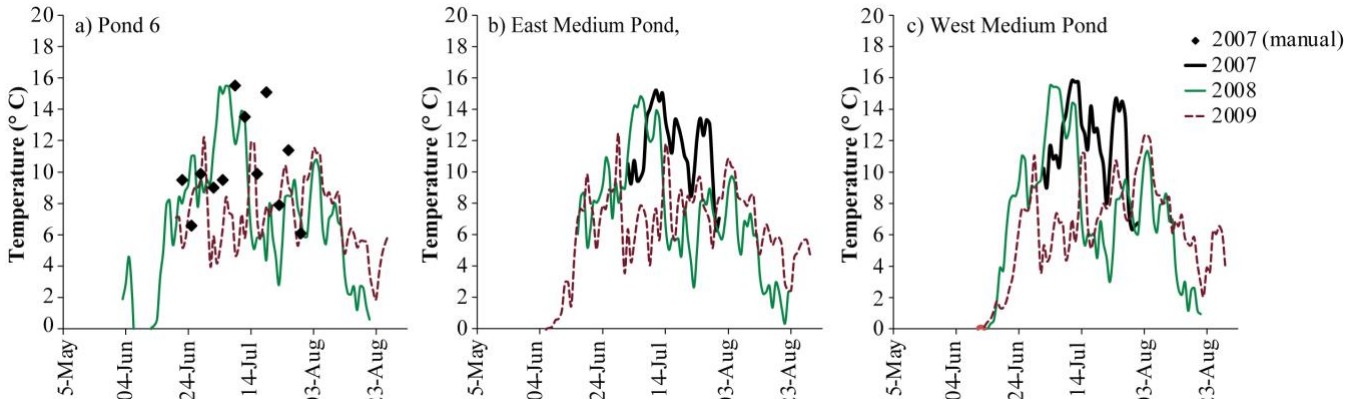

183

**Figure 4: Seasonal temperature regime of medium-sized ponds across Polar Bear Pass (2007-2009) based on manual (dashed line) and continuous data (solid line) from Pond 6 (a), East Medium Pond (b), and the West Medium Pond (c).**

186

## 4.2 July mean temperatures

Benyahaya et al. (2007) and others (e.g., Morison et al., 2023) indicate that water temperature is one of the most important parameters in ecosystem studies as it can influence both chemical and biological processes. Figure 5 plots the July mean temperatures of the central ponds at PBP, across the years. The July average temperature of ponds (8.5º C ±3.9) lying across the Canadian Arctic and boreal regions of Northern Canada with data collected from 1979-2009 are plotted for comparison (see Dranga et al., 2018). In addition, we also include July pond temperature data (11º C, 2005) from Eastwind Lake, Ellesmere Island (Woo and Guan, 2006) and from Cape Bounty, Melville Island (7.2º C, 2009 – Croft, 2013). Overall, the pond temperatures at PBP fall in the range found by these other Arctic studies, except it was much cooler in 2013 due to a late spring and cold summer.





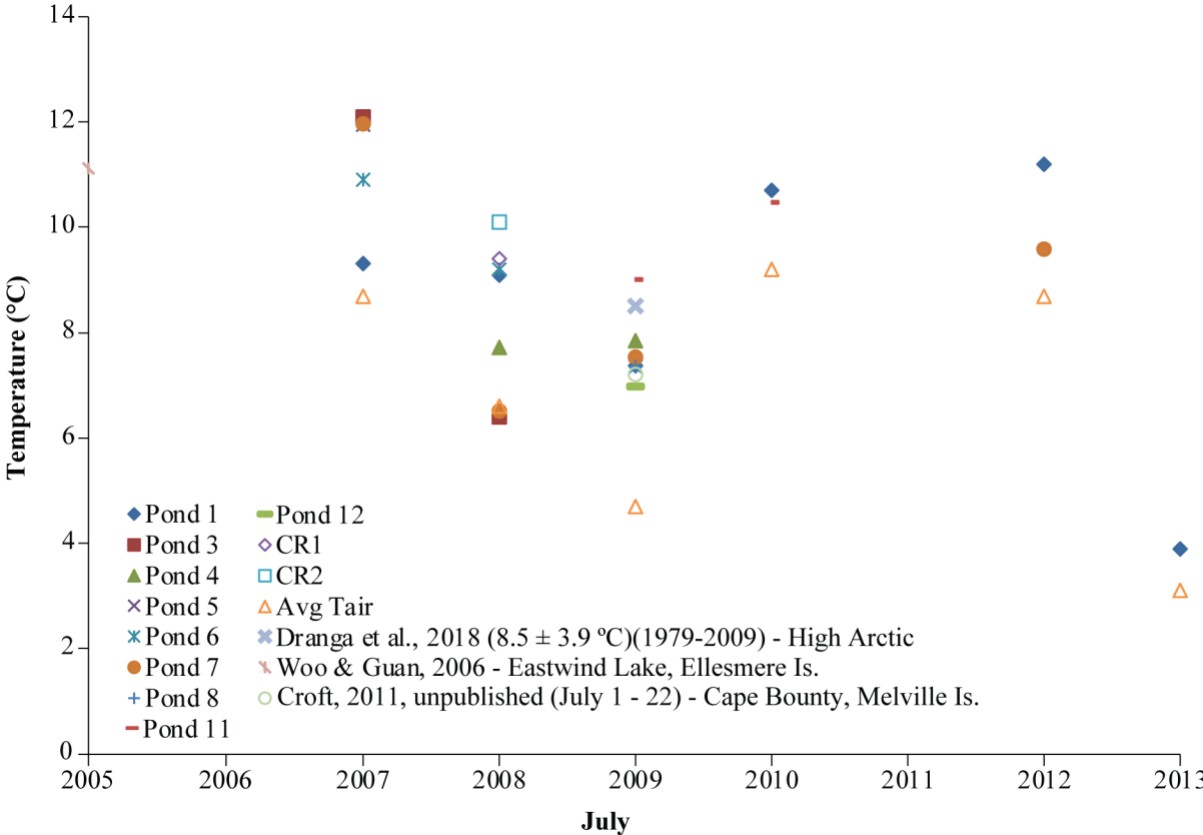

196

**Figure 5: Average July pond temperatures at PBP ponds. Pond data from other Arctic studies are included here for comparison. Illness in 2011 prevented pond temperatures from being obtained at PBP.**

**4.3 Cumulative relative frequency of pond water temperatures**

The cumulative relative frequency of Pond 1 from 2008 to 2015 based on continuous water temperature measurements is plotted in Figure 6. Comparable to Figure 2, there is considerable variability from year-to-year. In 2010, approximately 30% of Tw exceeded 20º C but in 2013, pond waters never reached this threshold. Only about 5% of the time did Tw reach 10º C. While 2012 was considered a warm season, Tw in Pond 1 only exceeded 15º C 20% of the time. It should be noted that the time frame of the study period in each year can impact these results. For instance, only 2008 and 2009 had a long record of pond water temperatures extending until the end of August. In 2008, Pond 1 exceeded 10º C about 20 % of the time. However, in 2009, Pond 1 was slightly warmer, >15º C, 20 % of the time.





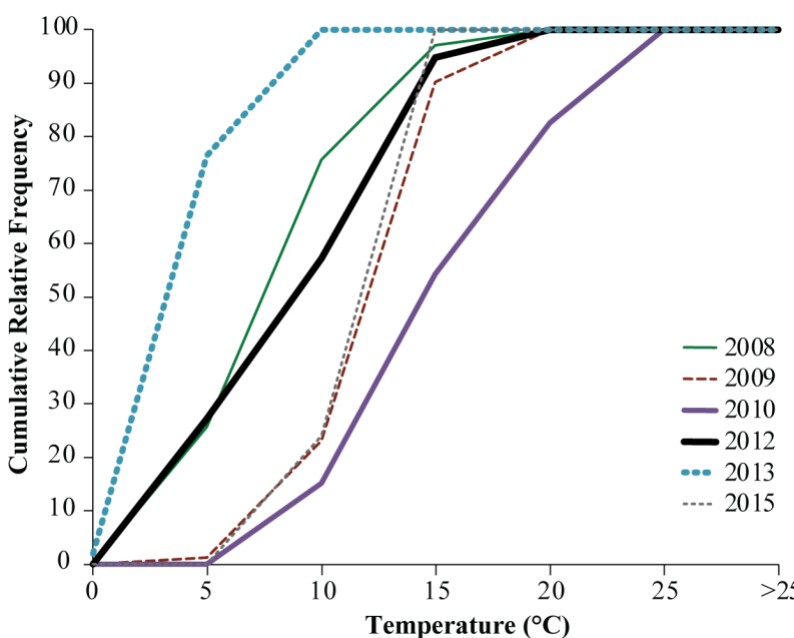


**Figure 6: Cumulative relative frequency of water temperatures (Tw) in Pond 1, 2008 to 2015.**


Figure 7, based on continuous daily mean data, shows the cumulative relative frequency of the central ponds. The pattern of
ponds in a year are similar to Pond 1 (see Fig. 6), but variability from year-to-year and occasionally from pond-to-pond does
exist (e.g., Ponds 2, 6). Differences in pond water depth, and water sources can impact water temperatures not just air
temperature. For instance, the Meadow Pond plot varies from the others, as it remains cooler for a longer duration. About 80%
of the time, the Tw here never exceeds 10ºC. As mentioned earlier, this is due to its location nestled in a small valley with
adjacent late-lying snowbeds that supply cool meltwater to it, for a protracted time (see Figure 8a, b). The Meadow Pond Tw
does not increase until after the late-lying snowbed disappears.





Figure 7: Cumulative relative frequencies of pond temperatures (Tw) of Central Ponds, 2007 to 2015.





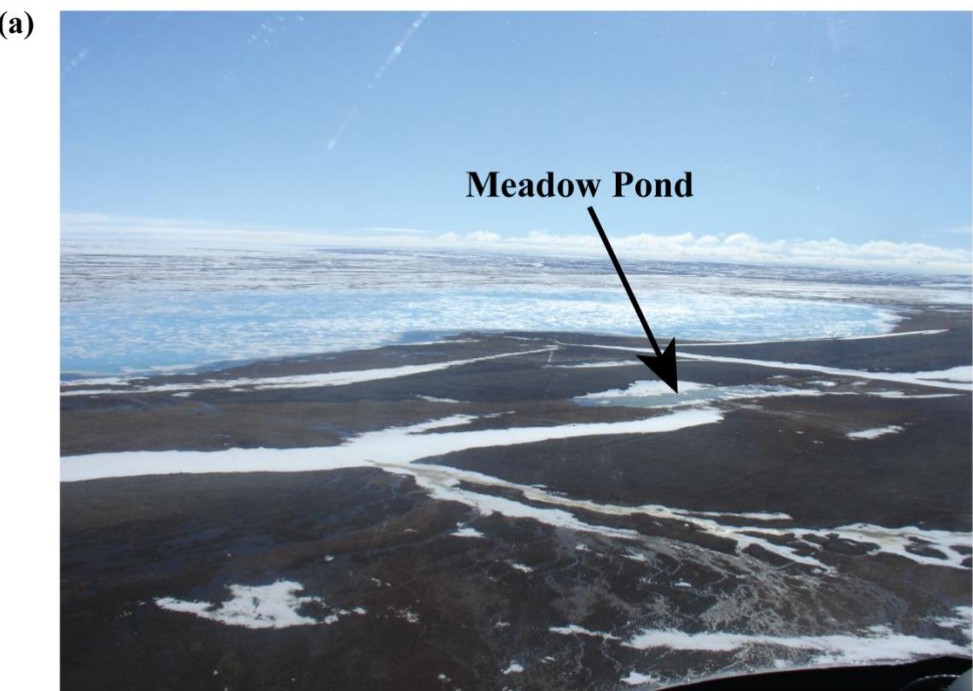

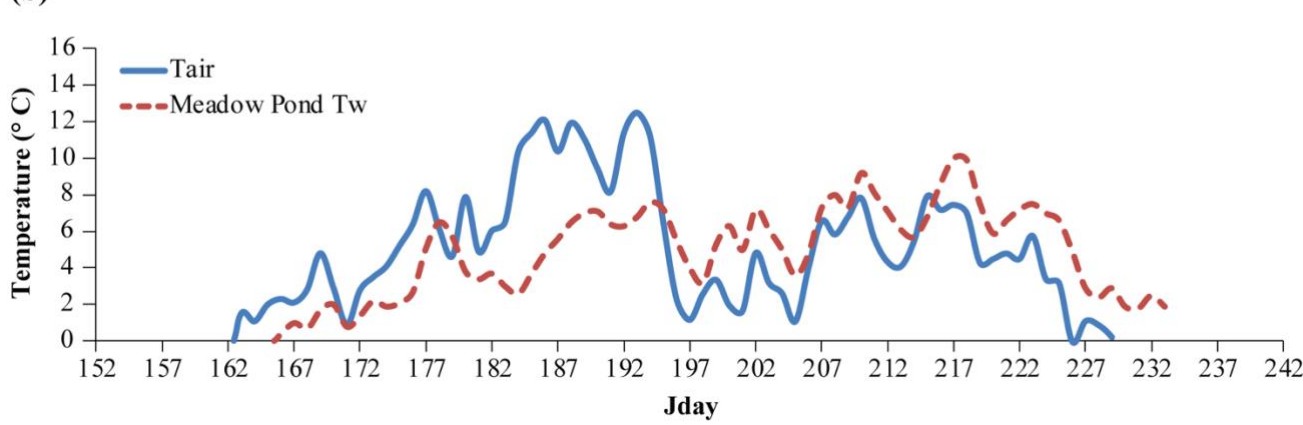

**Figure 8: Photo of the Meadow Pond (a) and air temperature versus pond water Tw (Meadow Pond) in 2008 (b). Note the late-lying snowbeds adjacent to Meadow Pond.**

Figure 9 shows the cumulative relative frequency of periphery ponds across PBP over several years. There is no significant difference in the pattern for these ponds, including size across PBP. The South Small Pond is shifted to the right in 2009 relative to other ponds, suggesting that it is warmer than the other ponds, but measurements did not start here until July 7 in 2009, which is generally the period when warmer air/water temperatures are reached. Steep temperature gradients can accelerate warming similar to rapid thawing and warming of frozen ground released later in the season (early July) from below



melting late-lying snowbeds (Woo and Young 2003). Given that the southern part of PBP (north-facing) is last to melt out
(Young et al. 2018), the ponds would generally lag in thawing and warming in comparison to the other ponds lying in the
northern part of PBP.

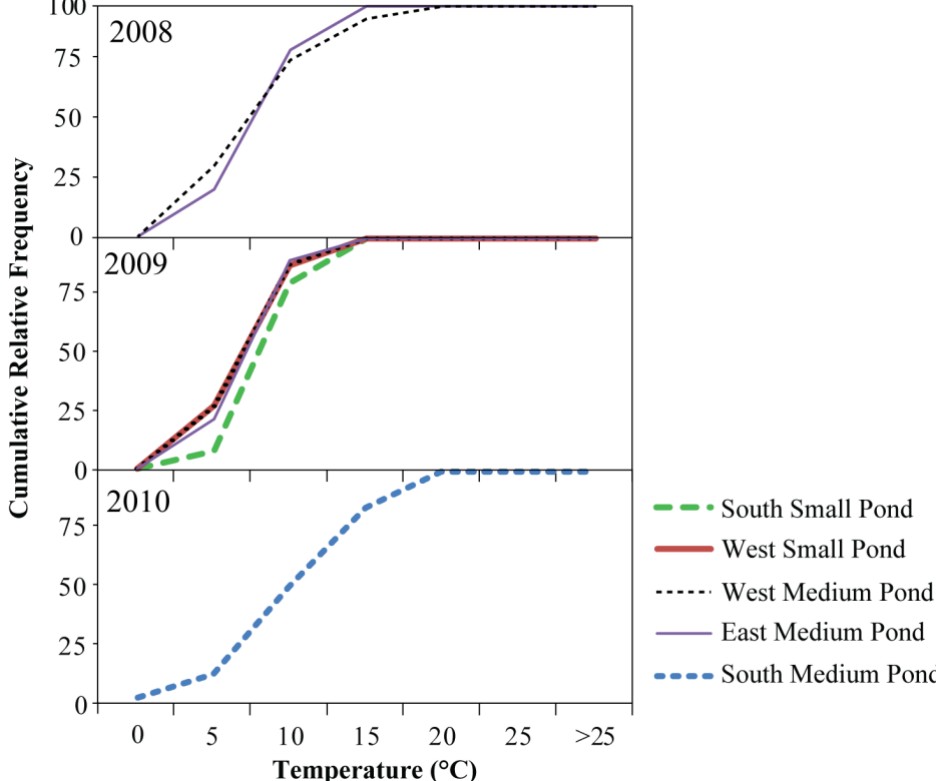


**Figure 9: Cumulative relative frequencies of pond temperatures (Tw) of periphery ponds at PBP, 2008 to 2010.**

**4.4 Air temperature and pond water temperature response**
Figure 10 provides an overview of the correlations between Plateau air temperatures and pond water temperatures in 2008 (a)
and 2009 (b); the two years when continuous measurements were made until the end of August (International Polar Years). In
2008, the relationship between air temperature and pond water temperature is strong, when temperatures are > 0° C. Here most
ponds show a correlation r > 0.8 or > 0.9, though pond temperatures are consistently warmer than the air temperature. As
mentioned earlier, the water temperature of Meadow Pond has a much weaker relationship with air temperature. This is due to
its location near melting late-lying snowbeds, which help to delay pond warming (see Figure 8).



a) 2008

b) 2009

**Figure 10: Correlations between Plateau daily air temperatures and Pond water temperatures (Tw > 0º C) for ponds 2008 (a) and 2009 (b), using continuous water temperature data. The black dashed line in the plots is the 1:1 line.**





## 4.4 Other environmental responses

### 4.4.1 Ground thaw

In 2007, it can be observed that there is considerable variation in pond thaw depth (0.3 m to > 0.7 m) (Figure 11a). Pond 5 had a mucky organic bottom, which might be indicative of high ice content, dampening ground thaw. Pond 11 exhibited a deep thaw > 0.6 m. It was a shallow pond with a rocky, blue-green algae (dark colour substrate), which would be effective in absorbing incoming radiation and accelerating ground thaw (Young and Woo, 2003; Young and Abnizova, 2011). Rapid thawing in the month of July can also be attributed to warm air temperatures (average Ta = 7.4º C).

Frost table measurements continued until the end of August in both 2008 and 2009 (see Figures 11b, c). Maximum frost table depths occurred around JD 210 in 2008 and slightly later in 2009 (JD 235). In 2008, the maximum depth reached in the ponds ranged from about 0.45 to 0.95 m, and in 2009, it was similar (0.42 to 0.95 m). The rate of frost table decline amongst ponds is similar in the early part of the 2008 season but then a wider range in thaw develops with the maximum spread emerging around JD 200 to 225 in 2008. The thaw pattern of ponds is similar in early 2009, but then pond thaw show variable depths by JD 210, which continues for the rest of the thaw period.

The well-defined freeze-back pattern is not as clear in 2009 as in 2008. While the rate and maximum frost table thaw in ponds might be different between the years owing to varying meteorological conditions, the thaw pattern of ponds is consistent; typically, Pond 5 exhibits a shallow thaw versus Pond 11, which can reach deep thaw, particularly in 2007 and 2009, and sporadically in 2008. This regular pattern in frost table depths amongst the ponds was replicated in 2010 as well (data not shown here).

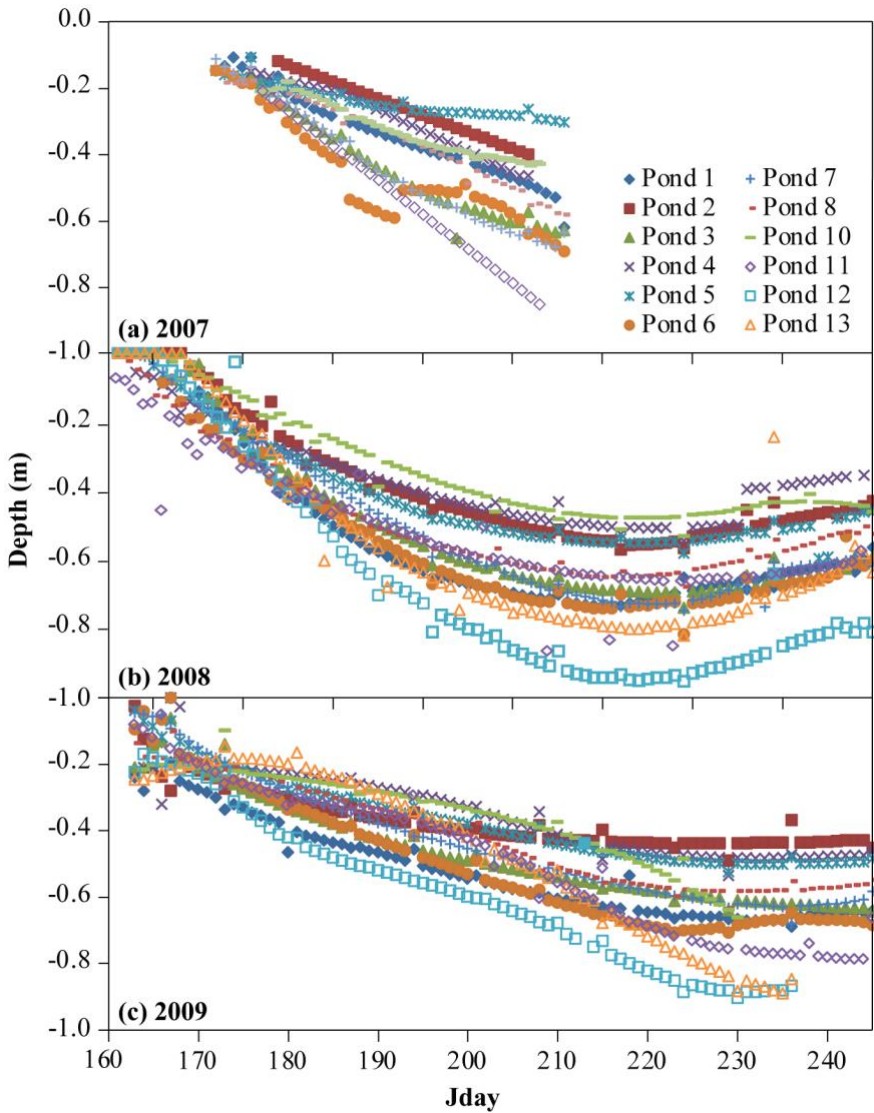

Figure 11: Frost table development in ponds at PBP in 2007 (a), 2008 (b) and 2009 (c).

**4.4.2 Pond water conductivity** (µS/cm)

Figure 12 provides the spot measurements of specific water conductivity of ponds over the 2008 and 2009 seasons. There is considerable variation between ponds throughout the 2008 season and the conductivity levels rise from ~100 µS/cm to most ponds exceeding 250 µS/cm (Figure 12a). Like water temperatures, meltwater from the nearby late-lying snowbed adjacent to the Meadow Pond helps to dilute it so the water conductivities are generally lower than other ponds. Pond 11, due to its rocky, blue-green substrate and deep ground thaw exhibits the highest water conductivity, maintaining levels higher > 400 µS/cm by early July 2008 (Figure 12a). A similar pattern emerges in 2009. Pond 11's water conductivity reaches > 400 µS/cm near the





end of June and about 500 uS/cm at the end of August. Pond 11 is an irregular shaped pond with a dark substrate. It was prone
to drying as was Pond 8. Like 2008, the Meadow Pond in early 2009 has a lower water conductivity signal than other ponds.
Later in the season, water conductivity levels start to rise at the Meadow Pond arising from the loss of meltwater from the
nearby late-lying snowbed (Fig. 12b).

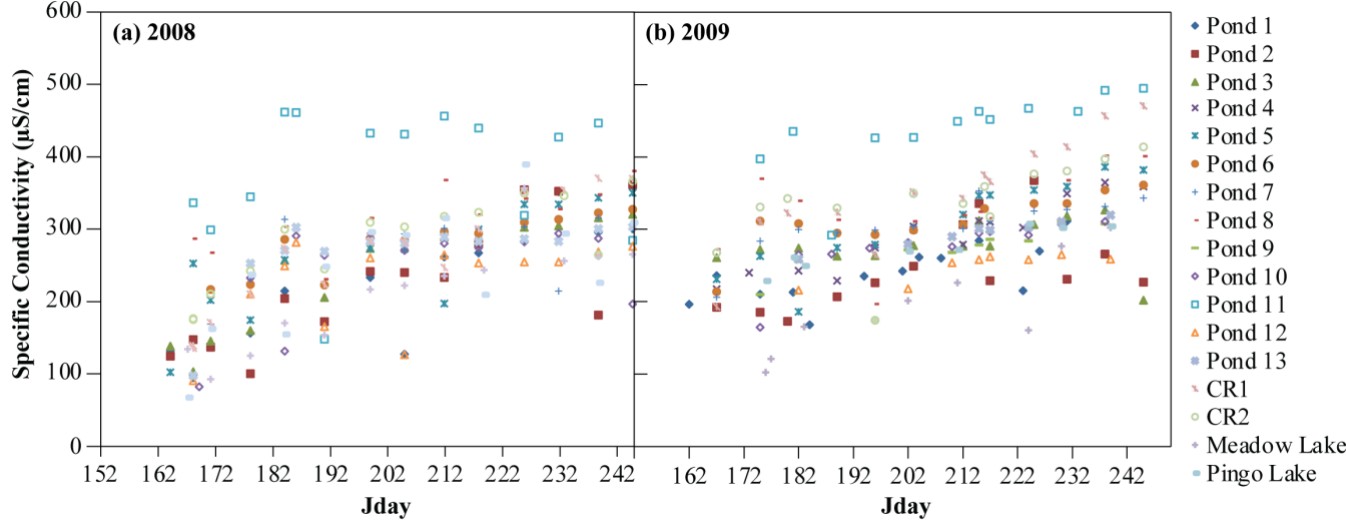


**Figure 12: Spot measurements of specific conductivity (µS/cm) of ponds at PBP in (a) 2008 and (b) 2009.**

279          Figure 13 provides an illustration of the impact of a warm season on the conductivity of pond water as measured by

a YSI 600 multiparameter sonde. Figure 13a, shows the hourly water temperature and conductivity of Pond 1, in the early
summer of 2008 versus the early summer in 2012. Pond temperatures do not vary that much in these two years during this
time, but conductivity only reaches about 250 uS/cm in 2008 (a cool season) but exceeds 400 uS/cm in 2012 (a warm season).





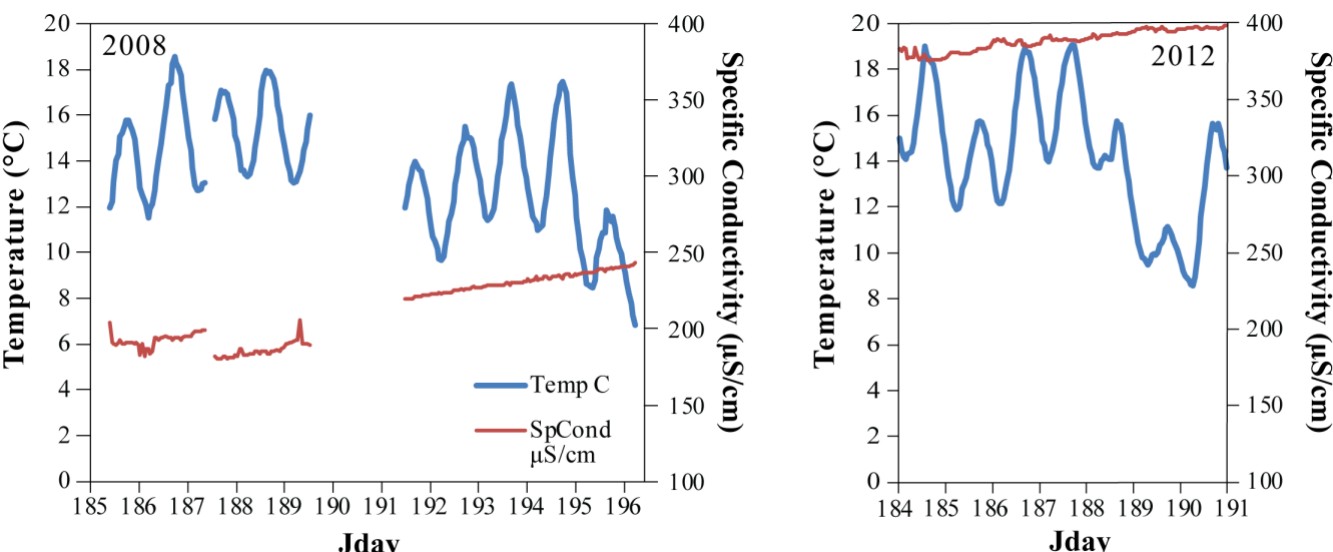

**Figure 13: Hourly estimates of specific conductivity (µS/cm) of Pond 1 in 2008 (a) and in 2012 (b).**

## 5 Discussion

### 5.1 Pond temperatures

McEwan and Butler (2018) examined air temperature and water temperatures in an Arctic pond over 40 years in Alaska's Arctic Coastal plain. They documented a 2.2° C increase over a 42-year period or roughly 0.5° C per decade. They found that the average thaw temperature of the pond increased during the first 30 days of the growing season from 1971 to 2012, and that the pond was warmer in the early spring. They found that temperatures in this study pond also correlated well across nearby and distant ponds (r= 0.93 – 0.99) suggesting that all the ponds extending over the coastal plain had undergone a significant change in their thermal dynamics over the past four decades.

At PBP, limited monitoring of the thermal regime of ponds indicates that most ponds show a rapid warming in the spring following snowmelt with most peaking in July. Then, pond water starts to cool in August. Average July temperatures are consistent with other pond studies across the Arctic; from the subarctic (Dangles et al., 2017) to polar oasis regions (Woo and Guan, 2006). Like McEwan and Butler's (2018) study, temperatures in central ponds all correlate well with each other and, for the most part, with ponds across Pass. This indicates that these ponds are all responding to similar climatic conditions (warming temperatures), which is also supported by strong correlations of pond temperatures with air temperature (> 0° C). However, there are still slight differences in the ponds owing to their unique settings. The Meadow Pond remains cooler than the other ponds owing to lingering meltwater from late-lying snowbeds, a pattern noted by others (e.g., Northern Ellesmere Island — Smol and Douglas, 2007; Somerset Island — Young and Abnizova, 2011). The South Ponds are delayed in warming relative to other ponds due to a regular and persistent snowpack - the southern part of the PBP has a north-facing aspect



ensuring later snowmelt and pond opening than the northern part of the PBP (Young et al., 2013; Young et al., 2018; Young,
305 2019).

**5.2 Environmental response to pond warming**

Lougheed et al. (2011) indicates the effects of warming and permafrost thaw on Arctic freshwater ecosystems remain poorly
understood. Generally, permafrost and the seasonal frost table maintain water levels in wetlands and ponds near the surface
but with an increase in thaw, the water table drops. As the permafrost thaws to deeper soil layers or is completely thawed, the
perched water table may be lowered, resulting in drier surface soils, and then this can lead to substantial carbon loss. But in
other areas, ground collapse can fill with water to form ponds and wetlands enhancing methane losses (Moonmaw et al., 2018;
Rehder et al., 2023).
Pond sediments depending on porosity characteristics can have varying ice contents (small to large). This can
influence pond thawing rates and maximum thaw depths. Permafrost degradation can also cause expansion or shrinkage of
wetland areas and warming associated with permafrost thaw could also turn wetlands into sources of carbon that increases
greenhouse gas emissions to the atmosphere (Moonmaw et al., 2018; Kreplin et al., 2021; Rehder et al., 2023). Wrona et al.
(2016) argue that while temperature is a key driver in ecological processes in tundra ecosystems, it is hydrological interactions
that mediate the climate responses of tundra ecosystems.
While pond temperatures were similar across PBP but varied due to climatic conditions (cool versus warm years),
there is considerable differences in the ground thaw rates, which can be attributed to the physical characteristics of the
sediments (coarser vs. finer), and colour. Some ponds had darker substrates than others, which can lead to greater absorption
of incoming solar radiation due to a lower albedo (Young and Abnizova, 2011). Warmer summers (e.g., 2007) versus cooler
ones (e.g., 2008) did lead to higher rates of pond evaporation (Young and Labine, 2010) but deeper thaw in warm, dry years
contributes to vertical pond seepage and drying. Pond thaw rates differ from pond-to-pond but they are consistent from one
year to the next confirming the critical role played by the pond sediments. A similar pattern emerges when evaluating pond
water conductivity. Ponds that are shallow and subject to drying tend to have the greatest water conductivities in both cool and
warm years. Here, shallow pond waters are in greater contact with thawing soil materials.
Roy-Leveillee and Burn (2017) studied the near-shore talik development beneath shallow water in expanding
thermokarst lakes, Old Crow Flats, Yukon. They found that near-shore taliks could develop in shallow lake/pond water, often
less than 20 cm when warm summers increased the thawing degree days. Deep and early snowpacks near the shoreline also
helped to keep lake bed temperatures above 0ºC, preventing permafrost aggradation. Roy-Leveillee and Burn (2017) argue
that "further work must include extensive examinations of permafrost sustainability in near shore and beneath the center of
shallow Arctic lakes in areas with varied climatic conditions, with particular attention to the increasing frequency of warm
years in circumpolar regions and to the effects of fluctuations in water levels resulting from changes in lake hydrological
regimes".





### 5.3 Climate warming: What can we expect for PBP ponds?

Climate warming over the last century has been greatest in the Arctic and is projected to continue in the 21[st] century at a rate
above the global average (Sim et al., 2019; Miner et al. 2022). Future hydrological changes, vegetation shifts and degradation
of permafrost have been identified as key areas of uncertainty in the prediction of permafrost carbon dynamics (Sim et al.,
2019). So, what can we expect for PBP? We will likely see continued warming, and if 2012 (a warm/dry season) is any
indication of future climate warming, we can expect early snowmelt, an earlier opening of ponds, a prolonged thaw season,
and extended periods of time when pond temperatures exceed 15° C. We can expect frost tables to thaw earlier in the season
and deeper, and that together with increases in evaporation loss, we may see pond water levels dropping below ground with
some some ponds drying out. This pattern is supported by Dyke and Sladen (2022), whose modelling efforts notes deeper talik
development under shallow ponds dotted on subarctic peat plateau landscapes.
It is likely that hillslope streams, which now provide some ponds with a source of water for an extended time during
the thaw season may not do so for as long (Young, 2019), and that ponds buffered now from elevated Tw and higher
conductivities by lingering meltwater contributions from late-lying snowbeds will cease to do so (Woo and Young, 2014). In
the high Andes, ponds and wet meadows are now being sustained by meltwaters from melting glaciers but they are predicted
to diminish and disappear by the end of the century. These ponds and wet meadows will then dry out and ultimately disappear
(Dangle et al., 2017). Moonmaw et al. (2018) also indicate that permafrost thaw can dramatically affect hydrology and that
fen-like systems are vulnerable as they rely on terrestrial water inputs. As these external water sources change or cease to
exist like the rapid loss of late-lying snowbeds at PBP and elsewhere across the Arctic islands (Woo and Young, 2014),
wetlands will be impacted. At PBP, many irregular shaped and shallow ponds (e.g., Ponds 8, 11) have already been observed
to dry out over summer seasons with vascular plants encroaching, suggesting that in the long-term, certain ponds may shift
into wet or dry meadow features (see Fig. 14), which may eventually alter greenhouse gas emissions (Rehder et al., 2023),
snowcover receipt, and evapotranspiration rates as shifts in vegetation have done elsewhere (Morison et al. 2023).
Strong correlations between air temperature and pond Tw will persist but departures will grow as pond water become
warmer than the air temperatures, especially shallow ponds with dark, rocky substrates (e.g., Pond 11). Lougheed et al. (2011)
observed that establishing a direct relationship between air and water temperatures is complicated by wind, water depth, ice
cover and other physical processes that determine temperature in small tundra ponds < 0.5 m at the International Biological
Program (IBP) site, Barrow, Alaska. They noted that variability in temperature is higher in both the early and late part of the
season, a pattern emerging for PBP ponds as well (see Figures 2, 3).





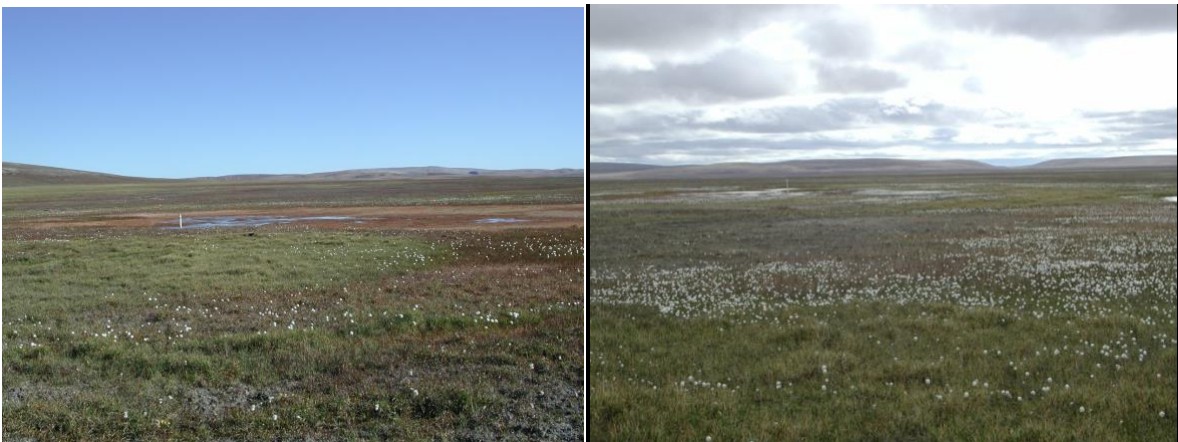

**Figure 14: Desiccated pond (Pond 8) in 2007 (left) and Pond 8, July 15, 2010 (right), showing how Eriophorum scheuchzeri -white cottongrass are encroaching.**

Linderholm et al. (2018) indicate that reanalysis data show an increasing trend in Arctic precipitation over the 20th century, but changes are not homogeneous across seasons or space. Possible increases in 21st century Arctic late autumn and winter precipitation counter increased pond evaporation losses to some extent. However, large uncertainties remain in predicting future precipitation patterns (Sim et al., 2019). If there are a series of years where fall rainfall fails to replenish pond storage deficits, and little spring snowmelt occurs, then it is likely that these ponds, especially the irregular shaped and shallow ones will dry out permanently, and vascular plants will encroach as has happened to patchy wetlands on Cornwallis Island (Woo and Young, 2014).

## 6 Conclusions

The thermal regime of High Arctic ponds was monitored over several years at Polar Bear Pass, a Ramsar wetland of international importance. Variability in pond temperatures occurs on a seasonal and inter-seasonal basis with ponds responding to warm and cool springs, and summer seasons. There is little variation in pond temperatures across the PBP but linkages to other terrestrial sources besides rainfall and seasonal snowmelt remain important (e.g., Meadow Pond). Prolonged inputs of meltwater from late-lying snowpacks into adjacent ponds serves to dilute and dampen water temperatures for extended periods in the summer.

Ground thaw is variable in ponds owing to substrate type, water depth and ground ice content. Future enhances in pond thaw may drain some shallow ponds, while others due to deep waters and abundant ground ice will be sustained. Like frost table patterns, the water chemistry (i.e., specific conductivity), which reflects evapo-concentration processes, groundwater flow varied amongst these ponds. The rapid and sizeable increases in water conductivity at Pond 1 in an extremely warm season (2012), highlight the large shifts in environmental responses that these ponds are now undergoing. Uncertainties





about fall precipitation exist for High Arctic regions. If pond storage deficits persist for several years and cannot be augmented by seasonal snowmelt or linkages to other water sources (late-lying snowbeds, streams), more ponds will disappear at PBP and may be replaced by larger wet/dry meadows. This will potentially have an impact on greenhouse gases and the feeding patterns of migratory birds.

**Data Availability**

The data that support the findings of this study are available at https://doi.org/10.5683/SP3/KGRQDO.

**Author Contributions**

Young carried out field work, analyzed the field data, and wrote the manuscript. Brown carried out field work, finalized diagrams, and helped to edit the manuscript.

**Competing interests**

Some authors are members of the guest editing team of the special issue in HESS.

**Acknowledgments**

We would like to acknowledge sabbatical funding from York U to K. Young in 2021, which allowed these data to be analysed and a paper to be prepared. Since 2006 to 2015, the Canadian Government Agency, Polar Continental Shelf Program (PCSP) has provided exceptional logistical support to Polar Bear Pass. We are also grateful to the many undergraduate and graduate students who have worked at Polar Bear Pass to help attain these environmental data. Jennifer Rausch, a wildlife ecologist with Environment Canada, kindly retrieved pond water temperature data in 2015. Finally, K. Young presented some of these findings at the 23[rd] Northern Research Basins Symposium and Workshop held in Northern Sweden, August 20-25, 2023.

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
