# Peer review of "Thermal Regime of High Arctic Tundra Ponds, Nanuit Itillinga (Polar"

_Hydrology and Earth System Sciences, 2023_

## Author Comment (AC1)

Replies to Reviewer #2

Thank you for your comments and suggests. They have helped to improve the manuscript.

In this paper the authors summarize arctic pond temperature and chemistry data to highlight their sensitivity to warming. There are some nice data presented, and the analysis is robust. The paper could be improved with the addition of a conceptual figure near the end that synthesizes the data. Such a figure would make the paper more citable as it would provide a hypothesis for future work about how these systems respond to climate warming. Superficially, Sometimes the English was a bit clunky and a good proof-read would fix this.

*Thank you. We have considered a conceptual figure at the end of the paper. We will have our paper edited to improve the grammar.*

My specific comments are:

Line 43: There is some off paragraph construction at this point in the paper.

*Yes, we have corrected in the text.*

Line 46: The authors juggle both Polar Bear Pass and Nanuit Itillinga at the beginning of the paper. I understand linking the two to avoid confusion with earlier scientific papers, but that only requires one instance. Perhaps pick one and go with that throughout. I suggest the Inuit name.

*We have decided to stay with Polar Bear Pass and PBP as the short-form, as this has been used in previous papers of this study site.*

Table 1: I don't understand why "water table" is used. I assume this means "pond depth". I follow maybe where the authors are going with this, but maybe since "pond depth" is more appropriate in this situation.

*We decided to use the general term – water level, and we have corrected this in the text.*

Line 112: This sentence "Less frequent ...." is an example of the clunky writing.

*Thank you, yes, we clarified the text.*

Figure 3: Maybe lower the x-axis labels to improve clarity.

*Yes, thank you, we have done this in the diagram.*

Line 176: "due to proximity" instead of "owing to nearness"

*Corrected in the text.*

Line 180: "but it was different than West Medium Pond ...."

*Corrected in the text.*

Figure 4: This figure would be more informative and better match the text if the panels were by year, each showing all three ponds.

*Thank you, yes, we have corrected this in the diagram.*

Line 204: Could you re-do the frequency analysis with a common period available each year to address this problem?

*Thank you, yes we have done this.*

Figure 9: Some simple statistics would help to objectively show these distributions are similar or different and this would help move the paper from a description of the data to an analysis of the data.

*Thank you, we have completed some additional analysis and added details to the text.*

Line 344: Are not water levels dropping below ground and ponds drying out the same thing?

*Not entirely. Evaporation can draw down water levels in ponds.*

Line 353: Conversely, as rainfall increases, maybe there will be more runoff during summer and late summer. A few sentences and references about this would balance the discussion. Some references to lean on might be:

Beel, C. R., et al. "Emerging dominance of summer rainfall driving High Arctic terrestrial-aquatic connectivity." Nature Communications 12.1 (2021): 1448.

Bintanja, R., & Andry, O. (2017). Towards a rain-dominated Arctic. Nature Climate Change, 7(4), 263-267.

*Thank you, we already refer to this in the paragraph prior to the conclusions.*

Line 369: Similarly, the discussion in this paragraph needs more attention than it gets. This is where the conceptual figure could come in. One that shows what warming and wetting (with a change in ppt phase) might do to these ponds.

*Thank you. Yes, we added a schematic to the discussion.*

Line 383: Maybe rephrase to "Increasingly warm and dry conditions may drain ....."

*Thank you. We revised the text.*

---

## Author Comment (AC2)

**General Comments:**

In this paper the authors present a case study of the thermal regime of ponds in the Canadian High Arctic. Their study site is Nanuit Itillinga, formerly Polar Bear Pass, Nanuvut, Canada, and the data spans almost a decade (2007 – 2015). The data presented includes seasonal data of pond temperatures, cumulative relative frequency of pond temperatures, pond water specific conductivity, and frost table depths. This study is an important addition to expanding our knowledge of year-to-year variability of pond temperatures in the Arctic. Studies like these are rare, and crucial to document to expand our knowledge about Arctic tundra ponds and the climate change impact on Arctic landscapes. I don't see any major problems with this paper. Below are some suggestions for improvement.

**Specific Comments:**

Line 54-63: I would like a couple of more sentences added to this section from information that is in Table 1. For example, "We studied X number of ponds, that ranged a surface area spanning from X-X".

*Thank you for your comments. We made Table 1 a Supplementary Table, and we added additional details to it. We also added more information about the ponds in the main text of the paper.*

Line 180: add "Medium" after "East".

*Thank you, we fixed this.*

Line 193: The Croft reference in the reference list is 2011, not 2013 as stated in the text.

*Thank you, we corrected this.*

Line 204-205: Wouldn't you want to compare this for a similar time period? I am not sure you can compare between years if you are not using the same time period. I might be wrong on that, so ignore this comment if that is the case.

*Thank you. We did this (compared across similar time frames) and have revised the diagrams (added box plots) and added some additional comments in the main text of the paper.*

Line 234-239: Any thoughts on why r values are lower in 2009 compared to 2008?

*Thank you for observing this. We really don't have a specific reason for these correlations being higher in 2008 than in 2009, and we have noted this in the paper.*

Line 261: Any reason why the data isn't shown in paper? I guess you don't have to show it, but it would be nice to see that year's data too.

*Thank you. We did not add it, as noted in the paper, the pattern was similar to other years.*

Line 282: What is the temperature difference between the "cool" and "warm" season? Any idea on why there is such a big difference in specific conductivity? Is it tied to the temperature difference?

*Thank you. I indicated the temperature difference in the paper, and added some additional information in the text.*

**Tables**

Table 1: This is a lot of information, and not much of this is mentioned in the text. It is a bit hard to envision how this will show up in print, but this table is currently three pages. Would this be better placed in the Supplemental? I am fine either way, so I will leave this to the editor. There is a parenthesis missing for the bulk density unit in year 2009 (Line 80-81). There are dates listed in some of the fields (Line 86-87) and not the others. Perhaps better to stay consistent throughout the table?

*Thank you, we have decided to move Table 1 to a Supplementary Table 1. I also clarified the wording around maximum frost table and frost table thaw during a specified period in the text. I also provided*

*information about the span of information for the different years in the Table title and in the text of the paper.*

**Figures**

Figure 1: It is a bit confusing with the naming "a, b, and c". There is no letter assigned to the top left figure. One option is changing top left to "a": "Location of the PBP catchment on Bathurst Island, Nunavut (a) with the red outlined area zoomed in and shown in (b)." and so forth. Can you also add a scale bar to some of these figures? I understand this might be difficult for c (seeing as it is a picture), but it should be possible for the other images. Could lat and long be added to at least one of these maps? Is it possible to include the pond locations and numbers (e.g., Pond 1, Pond 2, and so forth)? This would give a visual on where these ponds are located.

*Thank you. We have improved Figure 1, and I have added a photo, which indicates most of the central ponds.*

Figure 2: The different thickness of the lines makes this figure (and other figures in the manuscript) a bit difficult to read. Is the inset needed? Also, why is the y-axis in bold letters? This comment is for all figures.

*Thank you. We were initially following after HESS rules for our line diagrams (colour blindness) but will explore more options to improve readability. We have double-checked on the y-axis in bold letters.*

Figure 3: There is an overlap of 01-Jun with y axis. Remove? Add parenthesis for °C.

*Yes, thank you. We have corrected the Figure.*

Figure 4: The discussion in the text is about comparing the data for location and each year. Would it be better to split these figures into years (2007, 2008, and 2009) rather than location?

*Yes, thank you. We have rearranged the Figure by year.*

Figure 5: This figure is a bit difficult to decipher with the different symbols (too small?), and some seem to overlap. Is there any other

way this can be displayed? Maybe it is better presented in a table? In the legend, the Dranga et al. reference states it is from 1979-2009. I am only seeing one symbol in the graph for 2009. This might be because of an overlap of symbols. In the legend it says that the Croft, 2011 is unpublished. The reference shows this as a MSc thesis. So published?

*Thank you. We have corrected this diagram. We had to remove the Dranga et al. reference, as from reading the paper initially, we thought that they were specifying the average July pond temperature but they were only reporting the average pond temperature over that time period for all of their pond data 1979-2009 (see Table 3 – Dranga et al. 2017). We also corrected Croft 2011.*

Figure 8: Can you please add the date the picture (a) was taken? Also, you can add that to the graph in b. You have used dates in prior figures, should you use dates here as well to stay consistent? This same comment applies to the other figures using Jday.

*Yes, thank you. We have corrected the figure, put in the date for the photo and have added all of the dates to the diagrams instead of Jday.*

Figure 11: The y-axis are overlapping between a, b, and c (you can't see 0 in b and c). I suggest that you add spacing between figures. Do you have to have a negative sign for y-axis? I suggest removing the "-". Also, why are there symbols? It could be a lot cleaner if lines were used instead.

*Thank you. We have corrected the diagram, added some spacing and used lines for the frost tables instead of symbols.*

Figure 12: Similar comments as Figure 5. The symbols are small, and it is difficult to read this figure. Add space between figures because of overlapping symbols.

*Yes thank you. We have corrected the figure.*

Figure 13: remove units in legend. Units are in the y-axis. Instead of "Temp" and "SpCond" in legend, write "Temperature" and "Specific Conductivity". There is plenty of room to add that in the legend.

*Yes thank you. We have corrected the figure.*

---

## Author Comment (AC3)

**Replies to Reviewer #1**

*Thank you for your detailed comments and suggestions. They have greatly improved the manuscript.*

Line 12: drop "and whenever possible"

*Corrected in text*

Lines 43-53: combine with previous paragraph?

*Corrected in text*

Line 56: "...with focused pond studies in 2008 and 2009." – reword?

*Corrected in text*

Line 95: Drop "In this paper", not needed; also, why not just say "We describe" rather than "we focus on describing..."

*Corrected in text*

Line 97: "whereas" instead of "while".

*Corrected in text*

Line 99: "Substrate type varied across the ponds" is this within the pond, or the surrounding?

*Clarified and revised in in text*

Line 112: "Less frequent manual measurements while manual estimates were made at distant ponds" Not clear

*Clarified wording in text*

Line 122: "avoiding the flattening of data at low and high temperatures" not clear?

*Removed wording in text*

Line 125: "normality" or "normalcy"?

*Took out section along with Line 126 in text*

Line 126: "Given that autocorrelation did exist amongst the data (k-1) and is commonly found when comparing air to water temperatures (Johnson et al., 2014), no further work was carried out to develop a predictive model between air temperature and pond water." Not clear this is an unsolvable issue. The autocorrelation of data is not an issue, it is the residuals in regression analysis. But also, can you remove the autocorrelation, perhaps using first differences or some other way? In any event, if you are not developing a model, then just drop this sentence?

*Thank you. Took out section in text.*

Line 149: should this be in Discussion?

*Removed and placed in Discussion.*

Line 153: Do you mean that Pond 1 and others show similar seasonal cycles? Reword?

*Reworded text.*

Line 174: "Warming is comparable to that seen in Figure 2 ...", or something to that effect.

*Corrected in text.*

Line 178: Are these separate t-tests, and is there a multiple comparison problem?

*Yes, these are separate t-tests, there is no multiple comparison problems. We are only comparing two sample means at a time.*

Line 187: Mean July Temperature?

*Corrected in text.*

Line 188: Move to Discussion or Introduction.

*Thank you, corrected in text.*

Line 189-194. Drop sentence "Fig 5 plots...". Move sentence from 194-195 to beginning of paragraph and add (Fig 5) at end. Since you mention here in the text the details of other data added, the caption within Figure 5 could be reduced to simply Dranga et al; Woo and Guan, Croft 2011.

*Thank you, the text and diagram have been corrected.*

Line 223-224 could be changed to: There is no significance difference in the cumulative relative frequency of periphery ponds across PBP (Fig 9).

*Corrected in text*

Line 224: "The curve of South Small Pond..."

*Corrected in text.*

Line 234: "Figure 10 shows the ..." Also, on the graphs, in 2008, most points are above the 1:1 line, except those near the origin; when pond temperatures are less than ~2oC, air temperatures are greater. Is this interesting, or simply noise? It is only a couple of points, but happens in all but East Medium Pond. It seems much rarer in 2009, but still sometimes seen.

*Thank you, we have added additional text to describe the pattern, and refer the reader to Fig. 3. The points are due to delay in ice-off for different ponds.*

Line 259: do you mean Pond 12 rather than 11? Line 365: drop first sentence?

*No I meant pond 11, as we were referring to 3 years of data, Pond 12 only had two years of data. I did add that Pond 12 had the deepest thaw in 2008 and 2009.*

Line 288: Why not start with the next paragraph with your results, and then incorporate the results of this paragraph (lines 288-293) in relation to yours.

*Thank you, I revised the text.*

Line 307: Perhaps reword in the form: "The effects of warming and permafrost thaw on Arctic freshwater ecosystems remain poorly understood (Lougheed et al. 2011)."

*Corrected in text.*

Line 313: Reword to "The porosity of pond sediments depends on ice content" – is that what you mean?

*No, I clarified the text.*

Line 318: Is this sentence generally true, or only in relatively wet tundra systems? What about dry shrub tundra or polar desert?

*I revised the text.*

Line 324: Perhaps explain more "... vertical pond seepage ...", Do you mean "...deeper thaw in warm or dry years contributes to more downward seepage of water into the deeper active layer which leads to drying of the pond." – or change this for your meaning. And does it have to be both warm and dry or either?

*I added additional text in the paper.*

Line 325: "... from one year to the next in the same pond ..." And is it only colour or is it other aspects? Your supplemental #gure shows grain size but is not mentioned in the text. It seems this is useful data, based on your conclusions, so perhaps move the figure into the text (unless HESS includes Appendix as part of the pdf of the paper, but at the end).

*I added additional text in the paper about soil texture and referred to Supplemental Figure 1 and Table 1. HESS publishes appendices.*

Lines 328-330: combine into one sentence? "In the Old Crow Flats, Yukon, Roy-Leveillee and Burn (2017) found that near-shore taliks could develop in shallow (often less than 20 cm) lake/pond water, when warm summers increased the thawing degree days."

*Corrected in text.*

Lines 331-335: Not clear why you mention this. Needed?

*Removed this text.*

Line 345: "conclusion" rather than "pattern"

*Corrected in text.*

Line 353: what do you mean by "terrestrial water inputs"? Do you mean "streamflow or groundwater rather than precipitation"?

*Clarified text*

Line 403: obtain rather than attain?

*Thank you, corrected in text.*

Line 453: Is the Lehnherr reference correct; the title and author list seem wrong?

*Thank you. We corrected the reference.*

Although acceptable, they can be improved to make it easier for the reader to study the results more quickly. However, I think it would be up to the author to decide if they want to make the effort, depending on how comfortable they are with graphics programming.

*The figures are not created with graphics programming; most suggested edits will be made, while adhering the best we can to the journal standards for colour-blind accommodations.*

Fig 1: Lat-Lon lines may be useful on the upper-left map, at least. Also, the upper-left map is not labelled (a), (b) ...; is this intentional? The a-b-c is confusing, and I am not convinced these letters are needed; the red squares seem to be enough.

*Revising this figure with new pictures and will add coordinates to the overview figure.*

In the graphs: Why are some lines dotted and others solid? Why are some thicker than others? It would be much nicer to make all solid lines, different colours and thinner. (The thick lines may be good for presentations, but in a paper, thin lines would be sufficient; too much is hidden behind the thick lines). It would also help if you maintained the same colour for the ponds, (when plotting them together), or the years (if that is what is being compared) between graphs. The journal can weigh in here, but these "pastel" colours don't separate much, and

bolder colours may be easier to see? But maybe this will create a problem with making accessible graphs. It would also be nice if the x-axis were the same for all plots – ie the same time span (ie 15 May - 15 Sept?), as this would allow for easier comparison between graphs.

*The journal has rules to follow for colour blind accommodation: the lines need to be identifiable without colour, hence the varying thickness and colours, all of which appear distinct in greyscale. We will explore more options to help readability with colours. However, some varying thicknesses and styles may still be needed. Will ensure individual ponds have the same line style across figures. Some figures will be adjusted to better match axes; however, that will not be suitable for all.*

Fig 2: Is the inset graph needed - doesn't it just duplicate what is already on the graph? It would be nicer if the x-axis intersected the y-axis at -10, and maybe a horizontal, faint dotted line at 0oC if you wish

*Inset will be removed.*

Fig 3: As above. Here is an example where you could use thinner solid coloured lines for the Pond temperatures, and use a dotted line for the Air temperature, to more easily indicate the difference.

*Air temperature can be adjusted as suggested, however, 7 clearly distinguishable lines are required in greyscale, so some pattern/thickness variation will be required.*

Fig 4: It would help the reader if these were plotted the same as other - stacked on top of each other rather than 3 across. Vertically oriented x on the x axis is a pain to read.

*This figure has been stacked vertically as suggested and reformatted to facilitate annual comparison.*

Fig 5: Maybe make the points larger, although points would cover each other? But is there some other way to plot these? You can see year-to-year differences (cold vs warm) but distinguishing any one pond or seeing any relation to other properties of the pond (ie sediment colour) is difficult in the present form. And these symbols are a problem; for example Pond 6 could also be Pond 5 and 8 simply plotted on top of each other. Maybe a spaghetti plot would work? Or google to get some inspiration of other types of graph; many websites also include the necessary code (NOT bar graph, for example). What is CR1 and CR2? You may consider a light dotted horizontal line, or a light bar (including the s.e.) across the whole graph area at the appropriate value for the Dranga et al average, explaining it in the caption (if I am interpreting this correctly). What it the slash on the y- axis between 10 and 12? If it is from the Woo and Guan paper, then maybe the x-axis should go to 2004.

*We will explore options for this graph, however, no coding is being used. CR1 and CR2 are just ponds*, *named that way from previous study.*

Fig 7: This stacked panels, with 2 columns is another way to present data from, for example, Figure 2, with each year in a separate panel. In that way, you avoid the issue of colour, although it is true that it makes it more difficult to compare between years. However, if the x- and y-axes are the same in all 7 graphs, and a faint grid is used, it may work.

*We will explore options for this graph.*

Figure 8: Here you switch to Julian day. Is there a reason for the 2 different x-axes? This makes it difficult to compare between figures.

*Dates have been adjusted to calendar.*

Fig 11: Why using points instead of lines? This makes it hard to distinguish between lines, as they cover each other. It is also "ugly". On the y axis, it is not clear to which graph the -1.0 applies, so perhaps eliminate it, except on the bottom?

*Figure has been changed to line and panels separated slightly.*

Fig 12: As in Fig 5 above. Maybe spaghetti plots may work? Or some other kind? On the vertical line separating the two panels, there are some lines. Are these from 2008 or 2009? Some separation would be helpful. And again, it seems that you should either use Julian or calendar dates for all graphs.

*As above, dates have been changed to calendar.*

**Data Files**

Data are included as Excel files in a zip file. Within the zip, the files are labelled as "DRAFT". Does this mean they are not final? If they are, then perhaps rename. Also, could you pass through the files and make sure all columns are well labelled, with units and if needed, with extra lines to explain clearly what each sheet and row/column includes. For example, in File "Fig10...", sheet "2009 correlations" there are unlabelled columns. In the same File, sheet "2008 tair tw" there are unlabelled columns, and rows do not align. In File "Fig 9..." it is not clear what many of the columns are. This is a nice dataset, but it may be frustrating for people (or the authors in the future) to use it in the present form.

*These will be updated for final publication.*

---

## Author Comment (AC4)

Replies to Reviewer #4

*Thank you for taking the time to read and provide comments to our paper*.

This study conducts a detailed investigation of the thermal regimes of small tundra ponds in the Canadian High Arctic. Researchers collected important and valuable data, including sizes and hydrologic connections of ponds, to understand their variability. The research findings highlight the complexity of pond temperature dynamics and their broader implications for ground thaw, greenhouse gas emissions, and wildlife patterns. The study's focus on a specific region may limit its broader application, year-specific weather anomalies could skew long-term trend analysis, and uncertainties remain about the contributions of various water sources to the ponds.

Overall, this is an important work that contributes to our knowledge of the Arctic ponds. I will suggest it for publication after the authors address my comments and suggestions listed below.

1. I suggest adding a discussion on how the results from this study could be extrapolated to other regions. How is this study region geologically equivalent to other similar areas that have open water bodies?

   *You have raised an important point. It is often difficult with remote work in the Canadian High Arctic to find comparable sites, as local and climatic conditions can dominate the processes. Access to other sites can be problematic. In the paper, we have tried to compare our results with other studies including the North Slope of Alaska, Ellesmere Island, Somerset Island and Melville Island. We did insert a general schematic in the Discussion, which hopefully provides some ideas of how ponds in similar settings may respond in the future. This diagram also builds on our pond work on Somerset Island (see Young and Abnizova, 2011, Wetlands, 31, 535-549). In our conclusions we have also inserted an additional line which says*

   *"Finally, future research efforts should focus on the response of other large wetland areas in the Canadian High Arctic including Alison Inlet (southwest coast of Bathurst) and Truelove Lowland, Devon Island in order to investigate the response of ponds to future fluctuations in climate."*

   *The SW coast of Bathurst Island probably contains more ponds than PBP, but we have not been able to work there yet, and these ponds due to isostatic rebound are prone to drying, especially during warm/dry years. A fly over in late August 2011 showed many ponds containing salt crusts. On the other hand, the eastern coast of Melville Island is prone to marine transgression so coastal wetlands in this area will probably experience more flooding and expansion in the future. Comparison with Truelove Lowland would be interesting, as wetland work was done there in the*

*1970's. This area is also considered a "polar oasis" area like Eastwind Lake -Ellesmere Island, so the climate should be slightly warmer and drier than PBP.*

2. Clark et al. (2020) applied the LAKE model to study the thermal regimes at three Alaskan lakes in a continuous permafrost zone. One of the major findings was that snow depth and lake ice period substantially influence water temperatures. It would be interesting to know the length of ice season for these lakes and how snow depth and early/late ice melt could affect the overall temperatures in the ponds, specifically for a cold summer season in 2013. If the ice period can be extracted from MODIS or related products, it would be interesting and extremely valuable to combine that data with the present lake temperature data.

*Thank you. The Clark et al. (2020) LAKE model sounds very interesting. In our paper we are dealing with small ponds, so to consider lakes is beyond the scope of the paper. Also, once the snow disappears off the ponds, the ice cover melts out rapidly, within a few days and then pond waters start to warm. We can see this pattern in Figs. 2 and 3.  You are correct about lakes. Hunting Camp Lake can hold onto its ice cover especially in cool years (2013).*
       *We have also carried out considerable snow work in this area using both field measurements and remote sensing technology (e.g., Howell et al. 2012, Hydrol. Process. 26, 3477-3488; Assini et al. 2012, Hydrol Sci. J. 57, 738-755; Young et al. 2013, Hydrol. Res. 44, 2-20; Young et al. 2018, Arctic Sci., 4, 669-690).*

3. In addition, I suggest including a description of future measurement activities at the ponds, if any. Will other data (e.g. GHG fluxes) be measured at these ponds besides thermal data?

*Laura Brown is continuing her lake and climate studies at PBP. Anna Abnizova, a former PhD student did do work on greenhouse gases in 2008/2009 (see Abnizova et al. 2014, Ecohydrology, 7, 73-90).*

Clark et al.,  Thermal modeling of three lakes within the continuous permafrost zone in Alaska using the LAKE 2.0 model, Geosci. Model Dev., 15, 7421–7448, https://doi.org/10.5194/gmd-15-7421-2022, 2022

---

## Author Response (AR1)

Dear Editor,

Thank you for your feedback regarding on our paper, all revisions have been completed as outlined in our responses to each reviewer with the exception of one item for Reviewer #3: We indicated we would change Figure 10 from points to lines, however, we have reverted back to points for better suitability and visual appeal.

Thank you very much,
L. Brown